# Perspective of Health Care Professionals and Cancer Survivors on the Usage of Technology in Consultations

**DOI:** 10.3390/ijerph21040430

**Published:** 2024-04-02

**Authors:** Amalie Roland Dons, Henriette Emmy Ludwigs, Katrine Ringvig, Sine Rossen, Karen Trier, Lars Kayser

**Affiliations:** 1Department of Public Health, University of Copenhagen, 1172 København, Denmark; ghp730@alumni.ku.dk (A.R.D.); qzr687@alumni.ku.dk (H.E.L.); hfx117@alumni.ku.dk (K.R.); 2Copenhagen Centre for Cancer and Health, 2200 København, Denmark; kg58@kk.dk (S.R.); fn56@kk.dk (K.T.)

**Keywords:** patient reported outcomes (PROs), cancer survivors, health care professionals, technology, laptop, consultation, rehabilitation

## Abstract

This study explored the integration of technology in healthcare consultations between healthcare professionals (HCPs) and cancer survivors. The research aimed to understand how technological tools influence the dynamics and environment of cancer survivor rehabilitation consultations. The study used Actor-Network Theory (ANT) to analyze the effects of new technological actors in consultations and Invisible Work Theory to uncover hidden workflows associated with technology implementation. The study combined observations and in-depth interviews with HCPs and cancer survivors conducted in March to May 2022, and a follow-up group interview in November 2023. The study revealed that technology’s presence notably impacts the relationship between HCPs and cancer survivors, with HCPs expressing concerns that technology disrupts the consultation and challenges the relation. Over time, HCPs gradually began to use laptops during consultations to varying degrees, although the resistance to fully embracing technology persisted. This resistance is attributed to perceived pressure from management and a mismatch with established practices. The findings address the challenges in digital literacy and confidence among HCPs to facilitate the effective incorporation of technology and enhance the patient–clinician relationship. This research contributes to a deeper understanding of the interplay between digital health tools and patient–clinician relationships, highlighting the complexities and opportunities in digitizing healthcare consultations.

## 1. Introduction

The healthcare system is becoming more digitized, and digitally based solutions are used to address different challenges in healthcare, including access to patient information, prevention, quality assurance, efficiency, and patient involvement [1,2,3]. Technologies like laptops are increasingly used in patient consultations [4]. A consultation is no longer between patients and health care professionals (HCPs) but a meeting between the patient, HCPs, and technology. As digitalization and the wish for patient involvement have progressed [1,5], patient-reported outcomes (PROs) have become increasingly important in healthcare [6]. PROs are data about the patient’s state of health, including physical and mental health, symptoms, health-related quality of life, and functional level, that are reported directly by the patient [7]. PROs are a tool to increase patient involvement in treatment and rehabilitation [8] and is a basis for dialogue, joint decision-making, and treatment/rehabilitation planning [9]. In Denmark, the health authorities have an ambitious strategy for the development and implementation of PROs in both treatment (provided by hospitals) and rehabilitation (provided by municipalities) [8]. By digitizing PROs, healthcare providers can improve treatment, gather more accurate and real-time data, and improve patient care, empowerment, and face-to-face consultations [10]. In the consultations, PROs can contribute to the establishment of a strong rapport with patients and help to gather detailed information about their experiences and preferences [9,11].

The impact of digitalization on the interaction between the patient and the HCP during a consultation has not been extensively studied. It is crucial to understand this in consultations where it is important to establish a safe environment and build rapport that enables a holistic and person-centered assessment of needs, specifically in consultations regarding rehabilitation, as it can impact the success of the treatment and well-being of the individual [12]. According to WHO, the definition of rehabilitation is “a set of interventions designed to optimize functioning and reduce disability in individuals with health conditions in interaction with their environment” [13]. Consultations regarding rehabilitation may focus on private and sensitive inquiries like sexual function, mental health, and fatigue [11], which are softer factors and metrics without endpoints [14], but other subjects might also be covered in the consultation. Using a digital device can be challenging because it requires a nuanced understanding of the individual needs of each user (HCPs and patients), which involves delving into more delicate and profound topics. If users do not understand the advantages of using PROs, it can lead to resistance towards their adoption, hindering the potential benefits for patients and HCPs and creating barriers. One barrier could be that the patients have trouble filling out PROs or do not see their value [11]. Another barrier to using PROs in the consultation could be that the HCPs are afraid of seeming digitally incompetent, as they have to use a laptop to address PROs in the consultations [11]. Integrating digital PROs requires changes in documentation systems and processes to ensure that the collected data are effectively used to inform treatment decisions and improve patient care [15]. Using digital PROs in the consultation can challenge the workflow and performance as it becomes necessary to conduct the consultation in new ways [11,16]. PROs can be shown on a laptop or screen in consultations, allowing a graphical display and possibly creating a more interactive and engaging experience for the patient and HCP. The way HCPs engage with their patients has been transformed with new technology [17], which introduces new actors, such as laptops, into consultations. In this study, an actor is defined, according to the Actor-Network Theory (ANT), as a human or non-human entity that plays a role in shaping the interactions and relationships within a network [18,19]. Technology is, therefore, an important actor within the network concerning PROs.

Introducing new actors within the workflow of PROs can result in invisible work for the HCPs, as they have to maintain the value of using the PRO forms [14]. Invisible work refers to the work performed behind the scenes and effects of actors that may not be immediately visible or recognized [20]. Implementing PROs can require training and may add additional burdens to existing workflows [21]. It is essential to acknowledge and value the invisible work and be aware that specific societal structures and biases will render particular work invisible, leading to an underappreciation or lack of recognition for work that may be crucial [22]. It can be a tool to illustrate the work that the HCP performs using their experience and knowledge to support and keep the patients motivated to use technology [14]. When new actors are added or roles are changed, the sociotechnical approach of ANT can conceptualize how the dynamics change within the network [23]. ANT attempts to eliminate the division between human and non-human actors and emphasizes their role in shaping social phenomena [19]. It espouses a socio-technical symmetry and focuses on how actors influence one another, which affects their relation in a heterogeneous network formed by translation [18,23].

This research aimed to understand how technology influences the dynamics and environment of cancer survivor rehabilitation consultations. Our study investigated how HCPs use technology in consultations with cancer survivors. We explored the potential barriers and the impact of introducing technology into consultations using ANT to understand how new actors influence the dynamics and the invisible work to understand potential or hidden workflows that may occur. The objective of the study was to investigate the HCPs’ and cancer survivors’ perspectives on the usage of technology in consultations.

## 2. Materials and Methods

### 2.1. Study Design

The study is a qualitative study consisting of two parts (Figure 1): part I involved observations to understand the context and develop interview guides, followed by interviews with HCPs and cancer survivors. Part II consisted of a follow-up group interview based on the findings from the initial interview with HCPs and cancer survivors to gain knowledge about possible changes in the HCPs’ workflows since May 2022. As the context had become more digital in its workflows and a new digital actor had been introduced since part I, the group interview could also provide information about how new digital tools impacted the workflows in a given context and vice versa. This adds an additional dimension to ANT.

### 2.2. Setting

The study took place at the Copenhagen Centre for Cancer and Health (CCCH) located in Copenhagen, Denmark. Cancer survivors can be referred to the CCCH from the time of diagnosis and at any time point in the cancer trajectory, and rehabilitation programs are allocated based on a needs assessment consultation [24]. Around 1500 cancer survivors are referred to the CCCH yearly, and 22 HCPs at the CCCH complete the needs assessment consultations [25]. Since 2019, the CCCH has used PRO measures to support systematic needs identification at the initial consultation. In March 2023, the CCCH implemented a digital solution KommunalPRO (KPRO) that combines PROs with an algorithm to graphically display important symptoms, functions, and decision support at the individual level.

The hexagon in Figure 1 describes the interactions in the network at the CCCH. The cancer survivors are in the network as they are responsible for completing the PRO questionnaire and participating in their rehabilitation. The HCPs are responsible for the initial consultations at the CCCH and the cancer survivor’s rehabilitation. The digital PRO questionnaire is a form used in the consultation to obtain a snapshot of the cancer survivor’s health status (physical and mental health). A laptop is used in the HCPs’ workflow to access the answered PRO forms or the electronic care journal. The interior design represents how the consultation rooms are furnished and highlights possible challenges when working digitally. The TV screen was identified in part II and is therefore connected with blue lines. It is a wall-mounted TV screen located in the consultation room and is used by the HCPs to show information to the cancer survivor. The network is also related to external actors who are not directly involved in the rehabilitation at the CCCH. The hospital and the patient’s general practitioner are external actors as they provide the cancer survivor with medical treatment in parallel with their rehabilitation at the CCCH. The home and social care are users of the same electronic care record as the CCCH and are therefore included as external actors. These actors are connected via dotted lines to the network at the CCCH. The electronic care journal is a documentation system used in the municipality, where the HCPs in the CCCH have access to the cancer survivor’s data.

### 2.3. Participants and Data Collection

The participants were HCPs (dietitians, nurses, physio therapists, and occupational therapists) at the CCCH and cancer survivors referred to the CCCH with a booked initial consultation using PROs. The participants were recruited from March 2022 to May 2022 for part I and in November 2023 for part II. Individual and group interviews were conducted with those who agreed to participate (Figure 2). In part I, one HCP did not want to participate due to other related work tasks and one cancer survivor was reluctant to participate. We did not contact the cancer survivor due to ethical reasons. In part II, one HCP did not participate because of other work assignments.

The authors conducted all thirteen interviews and the group interview in Danish.

The interviews in part I were based on an interview guide for semi-structured interviews, which was informed by ANT and invisible work, and the interview guide for the group interview was constructed to triangulate the findings in part I.

The subjects for the HCPs included their preparation for the consultation with the cancer survivor; the interior design of the consultation rooms and the interactions with laptops, including the use of body language; the use of IT in their private life; and the workflow during COVID-19.

The subjects for the cancer survivors included their experience in filling out the PRO form; how it was used in the consultation; the cancer survivors’ opinion of using a laptop in the consultation; and the HCPs’ body language.

The individual interviews were scheduled for 30 min, but lasted between 15 and 25 min. The group interview was scheduled for 60 min and lasted 50 min. The participants were not offered a chance to review the transcripts or data after data collection.

### 2.4. Quantitative Assessment of Self-Rated Digital Competence

The HCPs and cancer survivors were asked to rate themselves on a scale from 1 to 5 based on Dreyfuss and Deyrfuss [26,27]’s theory about how individuals’ skills range from novice to expert. The item was formulated as “If a colleague should rate your level of skills, how would they then rate you in relation to your professional use of it, on a scale of 1 (novice)-5 (expert)?” [26,27]. The cancer survivors were asked, “If a friend or relative should describe your level of computer skills in their everyday life, how would they then rate you on a scale from 1 (novice)-5 (expert)?”. The scores were used to understand each participant’s level of Information and Communication Technology (ICT) competence (see Table 1).

### 2.5. Data Analyses

Prior to the analysis, a codebook was constructed in relation to ANT and invisible work that contained seven categories. The method used for transcribing, coding, and analyzing the interviews was abductive [28]. Initially, a codebook was constructed based on ANT and invisible work with seven categories, 17 sub-categories, and 55 initial codes. The coding of the first interview in part I was discussed between the authors to align the codes with the seven initial categories. Here, 2 codes were added. For part II, the data from the group interview were categorized into the 17 sub-categories from part I, and five initial codes were added to follow up on the initial data and gain knowledge about any changes. The coding was performed individually and reviewed by the authors. The software program Nvivo 12 (part I)/14 (part II) (QSR International, Burlington, MA, USA) was used to organize and categorize the data. Nvivo software is available for qualitative data analyses [29].

All interviews were conducted in Danish and all coding and analysis were conducted in our native language. For presentation in the “Section 3”, translations were performed by the authors ARD, HEL, and KR and was revised by the author SR who is proficient in English, ensuring that the meaning was intact. Quotes are used as examples to demonstrate our analysis.

The 13 interviews (part I) were analyzed based on the theoretical framework of ANT and invisible work. Between part I and II, an analysis was conducted. After reviewing the results from part I, it was decided to conduct an additional group interview (part II) to gain knowledge about any changes since part I.

### 2.6. Ethical Considerations

The participants were given written information about the aim of the study, the researchers, and the collection and handling of data. They were also informed that the participation was anonymous and that they could withdraw their consent at any time. Informed written and oral consent was obtained before the interviews and documented while recording. As no biological material was used in the study, there was no need for approval or exemption from the Danish National Center for Ethics, according to Danish legislation [30]. The interviews were recorded, transcribed, encrypted, and stored on a safe drive provided by the University of Copenhagen in accordance with Danish legislation (General Data Protection Regulation).

## 3. Results

The results of the self-rated digital competence conducted in part I are shown in Table 1. None of the participants (HCPs or cancer survivors) rated themselves below 3.

The same HCPs participated in part I and II; however, HCP 4 did not attend part II. There were three physiotherapists, one nurse, one dietician, and two occupational therapists. The age range for the HCPs was between 26 and 64; two were men and six were women. The age range for the cancer survivors was 49 to 68 years; three were men and two were women.

### 3.1. Part I: Initial Interviews

From the interviews, four categories emerged: resistance to laptops; laptops are necessary tools; laptops are acceptable tools; and laptops are a barrier. The categories help to shed light on how the cancer survivor enters the network described in Figure 1, and how the different dynamics in the network should be considered as an equilibrium. The cancer survivors should be considered as a group, with the technology which in this relation is the tools or interfaces used to interact with the cancer survivors i.e., the paper-based PRO-forms, but also the digital PRO forms. New changes in the interior design should also considered. The different actions and events occurring in the network were analyzed with reference to the terms translation, symmetry, and relation, which are used in ANT. We found that the changing roles and need for competence in these relations require the HCPs to develop new skills or workflows, which can become invisible.

#### 3.1.1. Resistance to Laptops

Cancer survivors’ resistance to using laptops during the consultation was primarily due to their experience with their treatment at the hospital.


*“[…] I’ve been to hospitals and the like in connection with the disease […] they just sit with the computer and write while they talk to me, so I just feel like I’m talking into thin air”*
—Cancer Survivor 10


*“[…] I have a general practitioner, and he seems very nearsighted, he sits with his head buried in that screen”*
—Cancer Survivor 9


*“[…] in consultations […] it’s not clever, it’s not. It doesn’t bother me that much, but […] it makes a bad, yes a bad [contact]”*
—Cancer Survivor 9

All the cancer survivors alluded to the relationship between them and the HCP. The laptop creates a barrier to this relationship. Their argument for not wanting a laptop in the consultation is that they want the HCP to be more present.


*“[…] it is very nice that we sit and look each other in the eyes instead of just into the air. It is as if she is there for me and not to type on that machine.”*
—Cancer Survivor 10

The above statements show that the cancer survivors’ experience is that the laptop takes away the HCP’s attention as they perform the documentation, leaving the cancer survivor feeling disregarded. In conjunction with this and the HCP’s own experiences with what the cancer survivors have told them, it has led to many HCPs avoiding the use of laptops.


*“[…] there [are] some who suggest that this could be done, for example, and it might not be as crazy as we have always criticized some doctors for because they sit there with their heads buried in the medical record and look at the screen more than they look at the patients. The patients do not feel met […]”*
—HCP 1


*“[…] I experience many who say that when they have been to consultations elsewhere, that being met by a health professional who is constantly sitting and looking down at the screen is really difficult […]”*
—HCP 7

Another HCP was told by a cancer survivor that when they were diagnosed at the hospital, the HCP’s attention was drawn to the laptop.


*“[…] it was odd […] ‘I was told I had cancer and she just sat and looked into a screen’, that’s something you hear”*
—HCP 6

In general, both the cancer survivors and HCPs have experienced situations where the presence of a screen harms the relationship, making it an impersonal experience during a vulnerable situation. The experience with the hospital negatively affected the relation with laptops and created resistance to their usage during consultations. A new translation associated with the laptop was difficult to achieve because the negative experiences were ingrained in the relation between the cancer survivor and screens.

#### 3.1.2. Laptops Are Necessary Tools

One cancer survivor finds using the laptop necessary during consultations and sees it as a required working tool for the HCPs.


*“[…] it’s just a normal conversation where he uses some aids to prop himself up, I think that’s fine enough […] I don’t have any problems with it, I think it went smoothly, there’s no complaining at all”*
—Cancer Survivor 12

The cancer survivor could not see any problem using the laptop during the consultation. It is a tool for a better and more effective process during the treatment at the CCCH. Even if the HCPs chose not to use the laptop in the consultation, they could see the possible value it can provide.


*“[…] [I] am also beginning to be able to use it as I really think something like this should be used, namely as a tool, as a help, so that I can do my work faster and easier and better. So that it eventually benefits the patients. […] it might be an advantage to have your computer with you so that you can go in and out of the stories [journal history]”*
—HCP 1


*“[…] it requires that you have sized each other up and does the other one think that you are not present enough, or do you have to point out, ‘I hear what you are saying, I am just writing at the same time.”*
—HCP 5

There is value and an advantage to using the laptop; it helps the HCP to be conversant with historical data, making the work more efficient. The HCP needs to understand the mood and feelings of the cancer survivors they are engaging with. Although laptops have many benefits and provide a workflow that can be more efficient, they can also create invisible workflows for the HCPs. The HCPs have to simultaneously interact with the laptop and cancer survivor while using their experience and knowledge to make them feel safe.

#### 3.1.3. Laptops Are Acceptable Tools

One of the cancer survivors was content with the laptop as a tool, but believed it is imperative that they are kept informed throughout the process.


*“[…] I would have liked it to be a little more visual, also in terms of sharing it with me and what it was (s)he saw and where, and why. Of course, it wasn’t difficult to understand that (s)he took point of departure in something I had answered, but it could be nice if (s)he had shown it more”*
—Cancer Survivor 13

The cancer survivor was not opposed to the use of the laptop but requested further visual involvement.

When asked whether the laptop was a barrier in the consultation, one cancer survivor did not see it as a barrier personally but expressed concern for other cancer survivors.


*“No, but you could well imagine that it could be for someone else. […] So, among other things, I saw someone at the reception yesterday who had to fill out that questionnaire on a computer and came in because she didn’t have a computer herself, and she looked like someone who was about to run the other way. She had never tried it before and stuff like that. […] So, I could well imagine that there were more people who would react differently to it, right?”*
—Cancer Survivor 11

The IT experience that the individual cancer survivor possesses will affect how the laptop will be met and whether it will create a barrier in the consultation. The interviewed cancer survivors are not necessarily against the usage of the laptop but wanted to be involved in the process.

#### 3.1.4. Laptops Are a Barrier

Another focus during the interviews was body language and eye contact. These are important elements of the HCP’s invisible work in showing presence and can be challenging when using a laptop in the consultation. The laptop can become a barrier as it can affect the HCP’s workflow.


*“Well, it’s all about presence. Simply, […] that I have my full attention on the people sitting in front of me.”*
—HCP 7


*“[…] I think about the absence of it [presence], and therefore I don’t want to bring the computer in […]”*
—HCP 8


*“[…] I don’t really think I should use it more, […] I would be able to document at the same time […]. I think it’s a bad idea to sit and write at the same time as I’m talking to someone, because it will affect the contact. […] our first contact […] is also about relationship building.”*
—HCP 4

The HCP’s main focus is on being mentally present and building rapport with the cancer survivors, not the laptop, so they do not want to bring the laptop into the consultation. The topics in the consultations can be difficult and may lead to complex and sensitive consultations. The HCPs must show professionalism and not appear affected by these topics. The laptop can be a valuable tool to stay objective and withhold emotions.


*“[…] it can be a tool, if you are busy, using a computer, because you can distance yourself and make things quite concrete and such […]”*
—HCP 8

The laptop can sometimes intentionally be used as a tool for distance, both for handling difficult emotions and when busy, as the laptop can create an invisible wall between the cancer survivor and the HCP.

The resistance to the laptop that most of the cancer survivors and the HCPs reported was mainly due to earlier bad experiences. The laptop was viewed as an actor that creates a barrier to the relationship during the consultation. The HCPs’ opinions were confirmed when they discovered the cancer survivors’ opinions. This process is an indicator of whether the laptop is a strong actor in the network.

### 3.2. Part II: Follow-Up Group Interview

Between part I and part II, new changes in the HCPs’ workflow occurred. We conducted a group interview to understand how the changes affected the network. We found that all the consultation rooms had a large TV screen to make it easier for the HCP and the cancer survivor to look at the screen together. The TV screens in the consultation rooms became a new actor in the network.

The following section represents the same categories as in part I (*resistance to laptops; laptops are necessary tools; laptops are acceptable tools;* and *laptops are a barrier*) to see how the dynamics and relation have changed since the last visit.

#### 3.2.1. Resistance to Laptops

The main reason for not using the laptop was the same as in part I. The HCPs are affected by the cancer survivors’ opinions and their experiences in other healthcare settings.


*“[…] I also have quite a few brain-damaged cancer survivors who get disturbed and see double […] they don’t want to look at a screen […] and if I need to print out an exercise program, there are quite a few senior cancer survivors who want it on paper […]”*
—HCP 1


*“[…] if there is a cancer survivor who is really upset during a consultation, then I would not say ‘now we just turn on PRO’. So, it is clearly situational and think of it as a tool that you can use if makes sense […]”*
—HCP 5

In some situations, the cancer survivors are not able to follow what is happening on the screen, either due to physical or cognitive conditions or emotional reasons. It would thus become a bad experience if the HCP uses the laptop in the consultation. Some cancer survivors preferred to not have the screen in the consultation and it was therefore not used.


*“I rarely use screens for consultations, […], I usually ask the cancer survivor if we should look at it together as the beginning of the conversation, most say no thanks.”*
—HCP 6


*“[…] it varies what takes up space and what is important [for the cancer survivor] in the consultation and therefore it would be an assault to force something like this into every consultation”*
—HCP 1

If the cancer survivor does not want the screen (laptop, TV screen) involved, the HCP prioritized this and did not bring the laptop in, as the cancer survivor was always the center of attention. This decision was also based on prior experiences and the CCCH’s view of presence and relationships in comparison to other health settings.


*“[…] It is […] again about the general practitioner who just sits and watches [the screen] or the doctor at the hospital who, at the consultation, has no idea who it is and the doctor is talking about. Some cancer survivors tell about experiences like that and they shouldn’t experience that.”*
—HCP 1


*“[…] But it is absolutely insane how many people a general practitioner or a specialist has to see during the day. They do not have time to notice nuances and so on if they open up that box [time for nuances], then you never come home, so it cannot be done. But we still have the time [for nuances], […]”*
—HCP 6


*“[…] We’re always talked about like it’s really nice to come in and meet a person here [CCCH] […]”*
—HCP 1

The HCPs in the CCCH are aware that cancer survivors are often met with a laptop in health settings. The HCPs at the CCCH understand and acknowledge the reality and the resources available in other health settings but they are also aware of the importance of this not being the case at the CCCH. Besides the HCPs’ own laptops, the newly introduced TV screen in the consultation rooms also creates problems in relation with the cancer survivor during the consultation. The TV screen is a disturbing element that is intimidating during the consultation, especially in smaller rooms. This changes the atmosphere in the room and the relation with the cancer survivor.


*“[…] So this whole thing about healing architecture, it’s almost disappears […]”*
—HCP 1


*“[…] In my eyes it’s a bit clumsy to put up something like that. I think there should be some really, really good reasons to put up boxes like that, and I haven’t really heard that yet.”*
—HCP 1


*“[…] As long as the layout is the way it is, I don’t think it can get smaller, and we’ve always known that the screens were too big, but that’s what we had the opportunity to buy […]“*
—HCP 6

The HCPs stated that the size of the screens negatively affected the *healing architecture.* Some HCPs required further evidence of benefits before approving the installation of the TV screen. As mentioned earlier, most HCPs would rather have smaller displays or none at all; however, the HCPs believed that the current PRO layout would not fit on a smaller display.

#### 3.2.2. Laptops Are Necessary Tools

The new actor changed the HCPs’ workflow and their approach to the screen during consultations. Compared to part I, all the HCPs now bring their laptop to consultations.


*“[…] it has changed for me so that I always bring a computer. I didn’t do that when I started. I went in and met the people sitting in front of me. I had a piece of paper with me and maybe the questionnaire they had brought with them. I always bring the computer with me now, so there is also more going on in the consultation in the way that I sit and book the rehabilitation interventions. I didn’t do that before, I went in and came back.”*
—HCP 7


*“I also didn’t bring the computer with me when I started here. I’m starting to do that now and I think it saves me time. Partly that and then I also think it’s nice that when you leave the consultation, you can say goodbye and thank you for today and then […] Go to your office. So, you end the consultation in the room, I like that, and I think it works well.”*
—HCP 8


*“[…] I agree with the thing about being able to finish in the room instead of finishing in the office and the [cancer survivor] is able to go home with an appointment instead of me having to call them with an appointment […].”*
—HCP 3

In particular, the possibility of booking a new appointment immediately with the cancer survivor has led to the HCPs now bringing their laptop. The HCPs have become better at explaining their actions on the laptop, and the workflow has thus become more transparent and better communicated.


*“[…] I think you’re more clear as to when we’re here [looking at each other] and when we’re on a screen, and it works really well, for me at least.”*
—HCP 5

The HCPs use their laptop as a tool to streamline practical tasks but are still able to distinguish between focusing directly on the cancer survivor and communicating when they must concentrate on the laptop.

#### 3.2.3. Laptops Are Acceptable Tools

A large part of the use and acceptance of laptops depended on the HCPs’ aptitude for IT. Some HCPs are uncomfortable and become stressed and feel *attacked* by having to use the laptop when the cancer survivor is present, leaving the HCP apprehensive and making their work less efficient and comfortable.


*“[…] I am stressing out about our system, the one I have to book in”*
—HCP 2


*“[…] Some days I think it almost feels like an assault that you have to use it and other days it is probably very good, […]”*
—HCP 1

The use of the TV screens in the consultation rooms was mainly based on the wishes of the management.


*“[…] just when they were implemented, the screens, I used it a lot because it was a desire from management that we should try to use them”*
—HCP 8

When the TV screens in the consultation rooms were implemented, the HCPs tried to include them in their workflow. The HCPs did not experience a positive change; instead, they experienced an increased workload.


*“[…] the times when I’ve […] had PRO up on the screen the way management wanted, the conversation takes much longer because you can’t overlook anything that’s on the screen. You have to like, relate to what is on the screen.”*
—HCP 7

The extra workload was caused by the need for the HCPs to go through everything shown on the screen. Sharing PROs on the TV screen removes the professional assessment of what is relevant to go through with the cancer survivor. Moreover, the use of the screen only occurred because of the management’s desire, not for the cancer survivor.


*“[…] I think you describe it very well HCP 8, the thing about really trying to use it, because I’ve also tried to use it to be skilled and a skilled employee and all that […], because for whose sake are we doing this? It’s certainly not for the person sitting across from [us].”*
—HCP 6


*“[…] I think many of us scabble along to try and hang on to, and remember everything that we are told to remember, because our focus is with the cancer survivor.”*
—HCP 1

The above has led to the screen not being used for PROs, and has instead been recognized for other practical purposes, such as education.


*“[…] If it’s something more specific, for example, I have a dietary guidance program where […] I can type in a diet plan or the like, and we’ll get it up on the screen […] then it’s educational.”*
—HCP 6


*“[…] I very much agree. I also use it often when I need to locate local exercise options [for the cancer survivor] now that you’re done here. Exercise guides where we use the ‘exercise map’ [webpage showing exercise options in the municipality of Copenhagen] and then it’s really easy to just run it up on the screen and then they can figure out how to use it at home […].”*
—HCP 8

The TV screen is used for another form of communication where the HCP educates or instructs the cancer survivors about their condition. This form uses more one-way communication than when the HCP and the cancer survivor go through the PRO form together.

#### 3.2.4. Laptops Are a Barrier

Ultimately, the HCPs stated that they have individual needs and ways of working. This applies to both the cancer survivors and the HCPs, which means that one solution will not suit everyone. The HCP–cancer survivor relationship and the resources provided by the CCCH are altered by using a laptop during the consultation as the focus in the consultation shifts from solely face-to-face interactions to incorporating technology and possibly becoming more efficient.


*“[…] So basically, I would be able to shorten a consultation a lot, if I thought that the PRO answers, was all that it was about. It could make the conversation shorter or much more result-oriented […] Then, you lose the feeling with the other human being, […] But we are able to do that (have longer consultations) because we have the resources, because we are lucky, we have to cherish that as long as we can, and if we can’t, then it is another service we deliver […]”*
—HCP 6


*“[…] You can’t come up with a solution that is a fit for all cancer survivors, or for all employees. We are as different as the cancer survivors […] it’s a good option, but not a ‘must’ thing. Our brains are vastly different, and I can’t pay attention to several things at the same time. For example, for me it’s disruptive and I deliver a poorer quality. It will be different for others, right? […]”*
—HCP 7


*“[…] it’s a lot about the fact that we’re very different people in here […] and I think there are some who are more skilled who can use it in a super sensible way and are damn good at it […] those computer things and […] they can just do something different with it than I can”*
—HCP 1

It is recognized that one solution does not suit everyone, as each individual has their own way of carrying out their work tasks. It is essential to be able to accommodate everyone to ultimately be able to provide the best service.

## 4. Discussion

### 4.1. Principle Findings

This study aimed to investigate how technology, specifically laptops, TV screens, and digital PRO forms, are used in consultations with cancer survivors and how it affects the interactions and physical environment. The essence is that the relationship between the HCP and the cancer survivor is affected when technology is used in the consultation. It is important for the HCPs that the cancer survivors do not feel that technology becomes an automated process and that the cancer survivors are a part of an assembly line as they have experienced in other health settings. Initially, the majority of HCPs did not use a laptop during consultations. At the one-and-a-half-year follow-up, and 12 months after the refurbished setting, all HCPs used their laptop for some purpose during the consultation, but not all of the HCPs were confident. This was probably due to many HCPs feeling that the implementation was forced into their workflow by the management and that the large wall-mounted TV screens do not match the interior design. The assessment of the self-rated digital competence in the initial interviews did not indicate a low digital competence for any of the participants. However, one contributing factor to a deficient adoption of the technologies may be an insufficient level of digital health literacy and confidence in technology amongst some HCPs. This also potentially results in uncertainty, which may be a stress factor in using technology when the cancer survivor is present.

### 4.2. The Socio-Technical Network

#### 4.2.1. Laptops as Part of the Relation

Laptops are sometimes intentionally used as a tool for distance, creating a barrier when handling difficult emotions and when busy. The screen was thus translated into a distancing tool, and had different properties attributed to it than intended. When the laptop was used as a distancing tool, it reinforced the negative experiences that the cancer survivors had in other health settings. The roles of the HCPs are tightly interwoven within the network. The technology should be seen as a part of this network rather than a technology in the organization. This will contribute to less friction and alignment for the performance of common tasks. It will also help to recognize how technology transforms and structures the current setting [31].

#### 4.2.2. The Translation of Laptops into Consultations

A key concept in ANT is the concept of translation, which is determined by negotiation and relation between the actors in the network [18]. A lack of interaction can destabilize the network, which can ultimately be abolished [18,19]. In part I, most HCPs did not use a laptop during consultations. As the interactions between the HCPs and the laptop were restricted, the laptop did not exist in the network during the consultation for most HCPs and, therefore, was a less critical actor. As stated by Berg (1999), because of the tight relations between the actors in the network, introducing a new element (such as laptops or TV screens) or the disappearance of an element (such as not using a laptop during the consultation) often causes reverberations throughout the work completed by the HCPs [31].

The HCPs in the CCCH believed that the use of laptops affects their presence and use of body language, which becomes a barrier. The solution for this challenge may be to omit the laptop as it is not necessary to accomplish the consultation with it. Laptops have not been assimilated into the network dynamics. The translation process did not succeed as intended, as the HCPs have not been convinced to integrate laptops into the consultation. In another study that examined doctor–patient relationship in a three-way consultation, consisting of the HCP, patient, and technology, the authors found that the power dynamics between the HCP and patient were modified when a computer was involved. The HCP must take the patients’ behavior, whether they watch, ignore, or exclude the screen, into consideration when determining how to use the computer best [4]. The HCP’s usage of the laptop/TV screen in the CCCH thus depended on the cancer survivor. According to a scoping review by Crampton et al. in 2016, the laptop can negatively impact the communication between the patient and the clinician. It was found that information sharing was greater when paper journals were used compared to digital ones. The clinician can have difficulties segregating their attention between the laptop and the patient [32]. In this study, it was strenuous for the clinicians to shift between eye contact and attendance to the patient and having to type on the laptop [32]. This also applies to the HCPs we interviewed, which leads to a stress factor for some of the HCPs. This can result in not using the technology as the HCPs do not feel confident in using the laptop while the cancer survivor is present. This can be accommodated by providing training when implementing a new technology and identifying the HCPs with low digital literacy to reduce stress and anxiety [33]. It is essential that the HCP establishes a trusting relationship with the cancer survivor; other studies indicate an association between the patient’s trust in technology and their trust in the clinicians [11]. In order for the cancer survivors to trust the technology, it requires the HCPs to feel competent in building relationships while working on a laptop. Before implementing technology within the organization, it is necessary to understand the specific network that constitutes a healthcare practice [31].

#### 4.2.3. The Symmetry of the TV Screen

As stated by Crampton et al. (2016), screen sharing can enable the communication and education of patients. Sharing health data between the clinician and the patient can educate the patient on their disease. At the same time, it provides an opportunity to validate the correctness of the patient data, making the laptop a tool to prevent errors [32]. One year after implementing the TV screens, the HCPs now use a laptop during consultations. This indicates that laptops have now become a part of the network; however, the TV screens are only used for practical matters instead of supporting the dialogue around the PRO content. The TV screens were translated differently than initially intended by the management.

### 4.3. Invisible Work

In the preparation of and during the PRO-based consultation, much of the work is recognized and documented. However, the increased digitization of the environment and expected use of laptops and TV screens in the interaction require extra preparations to hide uncertainty. This may increase the invisible work. On the other hand, the HCP had another kind of invisible work when laptops were not regularly used, as they had to go back to their PC stations to make bookings and other arrangements, adding an additional task to their workflow. This highlights how specific tasks or efforts remain hidden or undervalued despite their significance [22].

The invisible work lies in ensuring that the PRO forms’ value is maintained. The HCPs have to use the PRO forms in the consultation to show the cancer survivor that it creates value in their rehabilitation and encourages them to fill out the PRO forms again. The above also aligns with Torenholt et al. (2020), who described that for many cancer patients, the use of PRO forms is a reassuring task and a tool for verification, as the use of PRO responses and the reviewing of data affect the next response, creating a feedback loop [14]. Adopting digital tools for specific processes and improving patient care may result in unintended increased workloads and complexity for the involved health care professionals [20]. Working digitally during the consultation affects the interaction between the cancer survivor and HCPs and provides more invisible work for HCPs as they have to communicate the actions on the screen, complicating the use of body language and eye contact. In this way, the invisible work may contribute to unrecognized equilibria in relations or disturb the symmetry, thereby influencing the dynamics and the user experiences. One example is how the HCPs’ work efforts may not be fully recognized by their colleagues and management, thereby disturbing the relations in the organization.

### 4.4. Strength and Limitations

A strength of this study is that it contains interviews conducted with HCPs as well as cancer survivors, and we were able to gain knowledge about the use of technology from both perspectives. The group interview was a strength in this study, as it allowed for a comprehensive understanding of the HCPs’ use of technology after the implementation of TV screens. Another strength is that the participants were recruited by the author SR, who had insight into users of the technology, i.e., laptops and TV screens, so that the recruited participants represented the population of interest. This is, however, also a limitation as the recruited participants might not have expressed their opinion about the use of technology and instead expressed what they thought the management wanted to hear. Despite that, this was not our experience during the interviews. Additionally, the recruited cancer survivors may have more resources and be more willing to participate, which may also result in a higher digital literacy and not be representative of the population. Another limitation of this study was that there were no follow-up interviews with the cancer survivors after the implementation of the TV screens, and the cancer survivors’ opinions on the use of TV screens were not included. The limited number of participants (both cancer survivors and healthcare professionals) may be considered a limitation. We have reflected upon this and found that, given the setting with the same procedures and all patients being cancer survivors and the healthcare professionals working together in teams, the very specific context and limited population require a lower number to achieve a sufficient amount of information according to Malterud [34]. It should be acknowledged that further studies in other contexts are needed to confirm our findings.

### 4.5. Perspective

This study highlights the importance of considering how and which technologies should be implemented in consultations with the patients to maximize the benefits. It is of high importance to gain knowledge about the context in which the technologies are being implemented. Additionally, the education of HCPs in using the specific technology in the intended context is necessary for them to understand how to use it in the most beneficial way. When technological solutions are developed for consultations, it is important to be aware that, by including these, there is a risk of creating an environment where the relational work cannot be supported; thus, it is essential to understand the impact of implementing new technologies and how these can become an aid instead of an obstacle.

## 5. Conclusions

This study delved into the utilization of technology, including laptops, TV screens, and digital PRO forms, in consultations between HCPs and cancer survivors. The findings highlight how the dynamics between the HCP and cancer survivor is affected when technology is integrated into the consultation. Significantly, the HCPs expressed that the use of technology disrupts the consultation and makes it difficult to maintain the relationship with the cancer survivor. Throughout the study, the HCPs began using laptops during the consultations to different extents. The resistance to fully adopting the technology may originate from perceived internal pressure from the management or a lack of alignment with the established workflows. A critical factor affecting the seamless implementation of the technology was the digital literacy and confidence of the HCPs. Addressing these challenges is crucial for incorporating technology into healthcare settings and enhancing the relationship between HCPs and cancer survivors.

## Figures and Tables

**Figure 1 ijerph-21-00430-f001:**
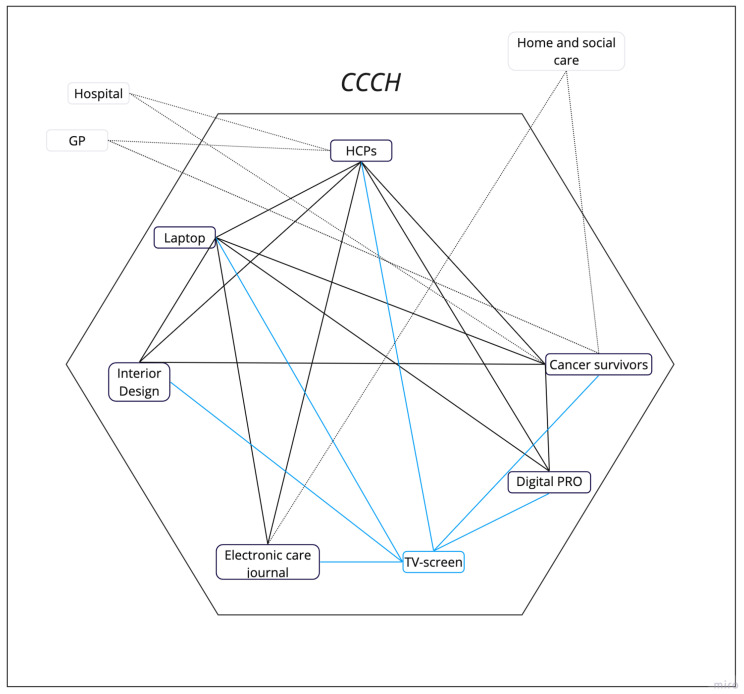
Illustration of the network of actors at the CCCH. Black lines: actors in part I; blue lines: actors in part II; dotted lines: external actors.

**Figure 2 ijerph-21-00430-f002:**
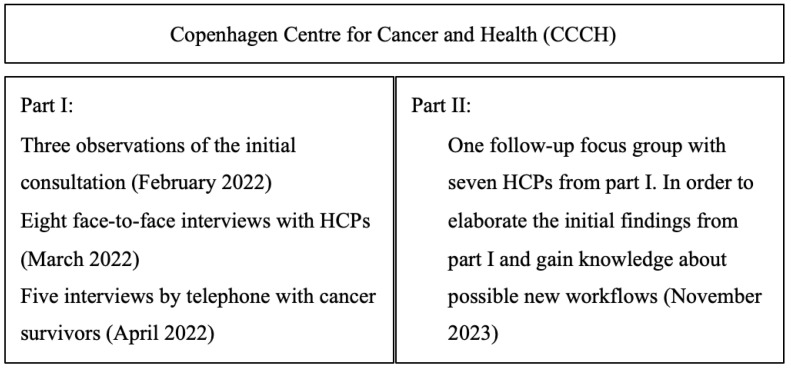
An overview of the study design and data collection.

**Table 1 ijerph-21-00430-t001:** Quantitative assessment of self-rated digital competence.

Participant	Self-Rated Digital Competence (1–5)
HCP 1	3
HCP 2	3
HCP 3	5
HCP 4	3
HCP 5	4
HCP 6	5
HCP 7	3–4
HCP 8	4
Cancer Survivor 1	4
Cancer Survivor 2	3.5
Cancer Survivor 3	4
Cancer Survivor 4	3.5
Cancer Survivor 5	4–5

## Data Availability

The data presented in this study are available on request from the corresponding author (due to privacy issues as the data are personal and health data and cannot be shared on a common platform).

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
