# Peer review of "Perspective of Health Care Professionals and Cancer Survivors on the Usage of Technology in Consultations"

_ijerph, 2024, doi:10.3390/ijerph21040430_

Round 1
Reviewer 1 Report
Comments and Suggestions for Authors
I would like to thank the authors for a very informative piece of literature. The manuscript is well organized and facilitates assimilation of information by the reader despite the length. I especially liked the usage of actual interview segments in brackets – this makes it easier for the outside people to relate to actual human feeling involved. A statistic simply does not provide that.
I believe the conclusions are well supported by the results presented and the discussion is relevant. In cancer care it is most often detrimental to patient care the implementation of additional changes (eg. technology, additional charts, additional consent forms, additional mandatory regulations) for two reasons and both have been discussed by the authors: firstly, the connection between healthcare professionals and patients is diminished (patients trust in a physician lacks when s(he) “googles” something while discussing treatment option (which has detrimental effects on cancer patients were trust is key), and secondly, the additional workload required from the healthcare professional can in fact be perceived as an assault and contribute to feelings of inappropriateness, depression and even complete burn-out. We should all consider, that in a world were treatment is centered around the patient and his feelings and experiences (which is by no measure wrong), the healthcare professional is also, a human being with feelings and experiences of his own that can, in fact, be subjected to increased pressure and negative workspace experience.
As a result, I believe the manuscript can be published after minor revisions consisting:
11. I do not understand what Ninvo is. I assume it is a software. Please add a definition and a reference source
22. Please remove authors contributions from the body of text – there is a “authors contribution” section at the end of manuscript
33. Please add as a limitation the fact translating Danish speech may result in losing some of the meaning
44. As a personal suggestion I would have added an quantitative analysis on how the implementation of technology affected the healthcare workers in terms of work overload, burnout, depression or other such manifestation and a quantitative analysis on how is affected the duration of actual health services provided to patient.
Comments on the Quality of English LanguageMinor editing needed
Reviewer 2 Report
Comments and Suggestions for Authors
This study investigates the integration of technology in healthcare consultations between HCPs and cancer survivors, aiming to understand its impact on consultation dynamics. Using ANT and Invisible Work Theory, the research combines observations and interviews to reveal that technology significantly affects the HCP-patient relationship. Over time, HCPs gradually adopted laptops, but resistance persists due to management pressure and mismatch with established practices. The findings highlight challenges in digital literacy and confidence among HCPs, contributing to a deeper understanding of the complexities and opportunities in digitizing healthcare. However, the article still needs to be improved in the following ways:
1. The sample size of the interviews is too small, and it is recommended to increase the total sample size.
2. P146-P152: It is recommended to compile the information of interviewees in a tabular format.
3. P218 results
In NVivo, sentences were constructed conceptually to extract regular concepts, and the initial concepts were continuously screened, summarized, and refined into relevant sub-categories. Based on the grounded coding process, relevant concepts were extracted and presented in a table.
Round 2
Reviewer 2 Report
Comments and Suggestions for Authors
The article in question presents a thorough and insightful study into the use of technology during healthcare consultations, specifically in the context of oncology rehabilitation. The use of both Actor-Network Theory and Invisible Work Theory to frame the research is commendable as it provides a multi-dimensional approach to understanding the complex interactions between humans and non-humans.
In summary, the research offers contributions to the understanding of digital transformations in healthcare consultations. It underscores the importance of acknowledging and addressing the human factors involved in technology integration, rather than assuming that technological solutions can be seamlessly inserted into clinical practice. The article effectively calls attention to the complexities of digitizing healthcare and the need for strategies that improve digital competencies and foster positive patient-clinician relationships in the face of such changes.